# Therapeutic Effects of Robotic-Exoskeleton-Assisted Gait Rehabilitation and Predictive Factors of Significant Improvements in Stroke Patients: A Randomized Controlled Trial

**DOI:** 10.3390/bioengineering10050585

**Published:** 2023-05-12

**Authors:** Yi-Heng Lee, Li-Wei Ko, Chiann-Yi Hsu, Yuan-Yang Cheng

**Affiliations:** 1Department of Physical Medicine and Rehabilitation, Taichung Veterans General Hospital, Taichung City 40705, Taiwan; paulleevghtc@vghtc.gov.tw; 2Department of Electronics and Electrical Engineering, Institute of Electrical and Control Engineering, Center for Intelligent Drug Systems and Smart Bio-devices (IDS2B) in College of Biological Science and Technology, National Yang Ming Chiao Tung University, Hsinchu 30010, Taiwan; 3Biostatistics Task Force, Taichung Veterans General Hospital, Taichung City 40705, Taiwan; 4School of Medicine, National Yang Ming Chiao Tung University, Taipei 11221, Taiwan; 5Intelligent Long Term Medical Care Research Center, Department of Post-Baccalaureate Medicine, College of Medicine, National Chung Hsing University, Taichung City 40227, Taiwan

**Keywords:** robotic rehabilitation, stroke, quality of life, walking speed

## Abstract

Robotic-exoskeleton-assisted gait rehabilitation improves lower limb strength and functions in post-stroke patients. However, the predicting factors of significant improvement are unclear. We recruited 38 post-stroke hemiparetic patients whose stroke onsets were <6 months. They were randomly assigned to two groups: a control group receiving a regular rehabilitation program, and an experimental group receiving in addition a robotic exoskeletal rehabilitation component. After 4 weeks of training, both groups showed significant improvement in the strength and functions of their lower limbs, as well as health-related quality of life. However, the experimental group showed significantly better improvement in the following aspects: knee flexion torque at 60°/s, 6 min walk test distance, and the mental subdomain and the total score on a 12-item Short Form Survey (SF-12). Further logistic regression analyses showed that robotic training was the best predictor of a greater improvement in both the 6 min walk test and the total score on the SF-12. In conclusion, robotic-exoskeleton-assisted gait rehabilitation improved lower limb strength, motor performance, walking speed, and quality of life in these stroke patients.

## 1. Introduction

According to the World Health Organization, stroke continues to rank second among the top 10 causes of death worldwide, behind only ischemic heart disease [1]. However, while stroke prevalence has increased, its mortality has actually decreased [2,3]. Therefore, many stroke survivors are left with post-stroke sequelae, such as pain syndromes, aphasia, dysphagia, depression, cognitive impairment, urinary incontinence, epilepsy, apraxia, neglect syndrome, and function impairment in upper and lower limbs. These post-stroke sequelae can cause long-term disability [4] and impose a great burden on their caregivers and families. Among various post-stroke sequelae, gait disturbance is the most concerning one for the patients [5]. Characteristics of post-stroke gait abnormality include hip hiking with leg circumduction, reduced foot clearance during swing phase, knee hyperextension during stance phase [6], and inadequate propulsion of the leg during pre-swing [7]. These gait abnormalities require that subjects expend more energy to walk and perform daily activities, leading to their frustration and depression [5,8].

Gait rehabilitation is therefore crucial for stroke survivors. To facilitate motor recovery, traditional approaches include neuro-developmental treatment [9], Brunnstrom movement therapy [10], proprioceptive neuromuscular facilitation [11], motor relearning programs [12], and the Rood method’s cutaneous stimulation technique [13]. These rehabilitation programs have been practiced by physical therapists for dozens of years. However, according to the landmark guidelines published by the American Heart Association/American Stroke Association in 2016, the therapeutic effects of these traditional approaches still cannot be established (Classification of recommendation IIb; Level of evidence B) [14]. On the contrary, it is highly recommended that post-stroke patients with gait limitations receive intensive and repetitive task training (Classification of recommendation I; Level of evidence A) [14], which is very physically demanding for therapists. Therefore, the duration of this highly helpful training technique is greatly dependent on the physical fitness of therapists. Hence, one recommended tool to deal with this problem is robot-assisted movement training according to the above-mentioned guidelines (Classification of recommendation IIb; Level of evidence A) [14]. Robotic-assisted gait training devices are attracting growing attention as they provide repetitive and intensive training while reducing the need for physical support by therapists [15]. Furthermore, some robotic devices can even accurately and objectively measure a patient’s physical performance and gait parameters during training. If combined with physiotherapy, these devices are believed to help more stroke survivors walk independently than those receiving only physiotherapy or standard care [16]. Recent evidence has suggested that patients in the first three months after a stroke, or those who cannot walk initially, benefit the most from robotic-assisted gait training [16].

Robotic-assisted gait training is categorized into exoskeleton and end-effector types, suitable respectively for different situations [17]. The exoskeleton type is used more frequently for patients with profound weakness, while the end-effector type is used more often for those with mild weakness [18]. End-effectors are attached to the distal parts of the extremities only, while the exoskeletons are attached to bilateral whole lower limbs [19]. The exoskeleton type is further divided into two subcategories: the treadmill-based exoskeleton robot and the orthotic exoskeleton. The treadmill-based robotic device allows movement training in merely one sagittal plane, which thus limits its therapeutic training effect on trunk balance. Patients can only be guided through a predetermined gait trajectory instead of walking volitionally. On the other hand, the orthotic exoskeleton allows patients to practice daily activities such as overground walking, sit to stand, stand to sit, and stairs climbing [19,20]. Nevertheless, a physical therapist needs to be involved more deeply during the training session in order to maintain the balance of the patient. The safety issue is thus more of a concern when using this type of robotic walking device. In this study, we used an orthotic exoskeleton for robotic-exoskeleton-assisted gait training.

Despite various studies that have been performed on the therapeutic effects of robotic gait training in the past [19], none have yet explored the predicting factors of significant improvement for those patients whose onset of stroke is within 6 months. The primary purpose of this study was to examine the effectiveness of robotic-exoskeleton-assisted gait training on the strength of lower limbs, walking speed, motor function performance, and quality of life in stroke rehabilitation. The second purpose was to determine predicting factors of significant improvement in post-stroke patients. We hypothesized that robotic-assisted gait training brings better strength recovery and functional improvement.

## 2. Materials and Methods

### 2.1. Participants and Trial Design

This was a randomized, controlled, prospective interventional study, which was approved by the ethics review committee of a tertiary medical center in Taiwan (No. CF21277A). The study protocol was registered at ClinicalTrials.gov (registration number NCT05825144). Each patient had signed an informed-consent form prior to participation. Participants, aged from 30 to 80 years, with hemiparesis due to either ischemic or hemorrhagic stroke in the last 6 months, were recruited in this study. The exclusion criteria included those with a significant history of pain or injury of lower limbs that affected their walking ability, those with cardiopulmonary diseases that prohibited exercise training, those with impaired cognitive function for understanding training orders or the questionnaire, those who were unable to complete the Timed Up-and-Go test, and those who were able to complete the Timed Up-and-Go test in <20 s using a walking device. The flowchart of our study is shown in Figure 1. Before the start of the rehabilitation program, we collected their basic characteristics, including age, sex, BMI (body mass index), lesion side of stroke, onset time of hemiparesis, score on initial National Institute of Health Stroke Scale (NIHSS), and serum levels of hemoglobin A1C, low-density lipoprotein (LDL), and homocysteine. According to our preliminary data from recruiting 12 patients, using the improvement of the time taken in the Timed Up-and-Go test as the primary assessed outcome, at least 36 participants needed to be recruited in our study to achieve a study power of 80% with a level of significance at 0.05.

### 2.2. Interventions

Patients were assigned to two groups using a simple randomization method. Specifically, patients fulfilling our study criteria were recruited by doctors during inpatient or outpatient visiting, and then the study assistant decided which group the participants would join by tossing a coin. Neither the patients nor the assessors were blinded. The control group received regular post-stroke rehabilitation programs 5 days a week, and the experimental group received robotic-exoskeleton-assisted gait rehabilitation 3 days a week and regular rehabilitation programs 2 days a week. The regular post-stroke rehabilitation programs were designed and performed by an experienced physical therapist, including facilitation techniques for the prominently weak motion of the affected limbs, strengthening exercise for specific muscle groups needed in daily activities, stretching exercise for the spastic muscle groups, trunk balance training, ambulation training, posture education, positioning advice, orthotic devices fitting, and functional training targeting improving the activities of daily life. As for robotic rehabilitation, we used the FREE Walk exoskeletal device (FREE Bionics, Hsin-Chu, Taiwan) to specifically address the issues of walking, standing up, and sitting down. Participants practiced the three kinds of movement repeatedly wearing the exoskeletal device on their lower limbs, with a study assistant standing behind to help the patient to maintain balance and control the training programs, as shown in Figure 2. Both groups were trained for 4 consecutive weeks, and the total training time for each day was the same across patients.

### 2.3. Assessments

The primary outcome measure of our study was improvement on the Timed Up-and-Go test, and the secondary outcome measures included improvement on the 6 min walk test, the quadriceps isokinetic muscle strength test using an isokinetic dynamometer (Biodex Multi-Joint System 3; Biodex Medical, New York, NY, USA), and the 12-item Short Form Survey (SF-12). These outcome measurements were conducted twice for each participant: once before and once after the 4-week post-stroke rehabilitation program. The purpose was to assess and compare the improvements in walking speed, function of lower limbs, quadriceps strength, and quality of life in the two groups.

The 6 min walk test is one of the most widely used clinical measurements of gait speed, function of lower limbs, and cardiopulmonary endurance. Its reliability in assessing stroke patients has been validated [21]. Participants were asked to walk for 6 min, back and forth, as fast as possible over a 30 m stretch of unimpeded walkway. Heart rate, oxygen saturation, blood pressure, rate of perceived exertion, and possible chest pain or breathing difficulty were recorded during the test. The total walked distance for each participant was the data we collected.

The Timed Up-and-Go test is also a common clinical tool to assess the composite motor function of lower limbs. It has proven reliability and validity, and it is an easy-to-administer measurement for patients after stroke [22]. For community-dwelling older people, the time taken in this test is significantly and independently associated with future falling events [23]. Since falling prevention is an important issue for stroke patients with gait instability, we selected this test as one of our outcome measures. Participants were asked to stand up from a seated position, walk for 3 m, turn around, and walk back to the chair before sitting down. The time needed to complete the test was our recorded data.

To measure the strength of lower limbs objectively, we used the Biodex isokinetic dynamometer to determine the peak torque force of the knee regarding flexion and extension at 60°/s and 120°/s. The test for each participant was performed by a single assistant and repeated three times. During the test, the assistant was to encourage the participant to push the lever arm as hard as possible. Only the maximum torque force among the three repeated tests was recorded. The isokinetic dynamometer is reportedly an effective way to test and train post-stroke patients [24]. Furthermore, it can better quantify the muscle strength of lower limbs compared to the most commonly used Medical Research Council (MRC) Scale in clinical practice, since the peak torque obtained by the isokinetic dynamometer is a continuous variable, while the MRC scale is just a categorical variable.

We further applied the 12-item Short Form Survey (SF-12) to measure the quality of life of individual participants. The SF-12 is a self-reported outcome measure based on answers to 12 questions. Briefly, these 12 questions are rating their health in general, evaluating health-related limitations on moderate activities and climbing several flights of stairs, evaluating physical health-related limitations on the kind and the amount of work or activities that can be performed, evaluating emotional-problem-related limitations on the amount of work or activities being performed as well as being less careful, assessing the impact of pain on working, rating the degree of feeling calm and peaceful, having lots of energy, and feeling depressed, and evaluating the impact of physical or emotional problems on social activities. The scores were converted into 8 sub-domains according to the following aspects: limitations in physical activities due to health problems, limitations in social activities due to physical or emotional problems, limitations in usual role activities due to physical health problems, bodily pain, general mental health (psychological distress and well-being), limitations in usual role activities due to emotional problems, vitality (energy and fatigue), and general health perceptions. The scores of the 8 sub-domains were further subgrouped into a physical domain score, a mental domain score, and a total SF-12 score. The SF-12 is shown to reproduce SF-36 summary scores for stroke patients with minimal loss of information [25].

### 2.4. Statistical Analyses

We used Predictive Analytics SoftWare (PASW Version 18.0, Chicago, IL, USA) to perform our statistical analyses. In the Kolmogorov–Smirnov test, our data were first found not to be normally distributed. Therefore, the Wilcoxon signed rank test was then used to determine statistically significant improvements after 4 weeks of training, regarding the following: peak torque force of the knee in 60 degree per second and 120 degree per second, the SF-12 total and its subdomain scores, the distance covered in the 6 min walk test, and the time taken in the Timed Up-and-Go test. We then used the independent sample T test to detect significant inter-group differences, if any, regarding the amount of improvement in these outcome measures. The effect size and mean difference of change were also calculated via this test. To analyze the predictors of better post-stroke rehabilitation progress, we divided the improvements in the SF-12 scores, the 6 min walk test, and the Timed Up-and-Go test of all patients into two groups using the median values as their respective boundaries. Next, we used logistic regression analysis on the baseline characteristics to determine the most predictive factor regarding improvement beyond median value. Statistical significance was set at *p* < 0.05.

## 3. Results

### 3.1. Demographic and Clinical Characteristics of Participants

Of the 120 participants screened for eligibility between January 2022 and December 2022, 82 of them were excluded based on our exclusion criteria (and they were referred to community hospitals for further rehabilitation). The remaining 38 were included for study. All of them had definite gender identity. A total of 22 of them were male, and 16 of them were female. Their average age was 67.42 ± 11.07 years old. Of them, 27 were diagnosed with ischemic stroke, and the other 11 with hemorrhagic stroke. They were randomly assigned to the two rehabilitation programs, and all completed the entire course. The experimental group had 17 participants, and the control group 21. Their demographic and clinical characteristics are shown in Table 1. We found no significant difference between the two groups.

### 3.2. Clinical Outcomes

#### 3.2.1. Within-Group Post-Rehabilitation Changes

Table 2 shows the results of both the control and experimental groups after 4 weeks of rehabilitation. In the experimental group, most outcomes showed significant improvement, except for the physical and mental subdomain scores on the SF-12. In the control group, significant improvements were found regarding knee extension and flexion torques at 60°/s, and the time needed to complete the Timed Up-and-Go test. On the other hand, there were significant reductions in the physical subdomain and the total score on the SF-12.

#### 3.2.2. Between-Group Comparison of Post-Rehabilitation Improvement

As shown in Table 3, the experimental group outperformed the control group in the knee flexion torque at 60°/s, the 6 MWT distance, and the mental subdomain and total scores on the SF-12. The largest effect size was observed in the SF-12 total score, while the smallest effect size was observed in the flexion torque of knee at 120 degrees per second.

#### 3.2.3. Predictors of Improvement Revealed by Univariate and Multivariate Logistic Analyses

Univariate logistic regression analyses determined which baseline characteristics were significant predictors of greater improvement in the following: the 6 min walk test, the Timed Up-and-Go test, and the SF-12 scores that exceeded their median values. As shown in Table 4, we found that robotic training was a significant predictor of progress in the 6 min walk test, and LDL cholesterol values were a significant predictor of progress in the Timed Up-and-Go test. Since only one variable was statistically significant in the univariate logistic regression, multivariate logistic regression analysis was not performed. Regarding the SF-12 in Table 5, robotic training, the initial mental subdomain score, and the initial total score on the SF-12 were significant predictors of progress in the SF-12 score. However, in further analysis with multivariate logistic regression, only robotic training was found to be significantly predictive.

## 4. Discussion

In our study on post-stroke patients, both the robotic-assisted gait training and the traditional rehabilitation programs were beneficial for strength recovery, the function of lower limbs, and health-related quality of life. However, only the robotic-assisted gait training brought significant improvement in the 6 min walk test. Furthermore, the extent of improvement was significantly greater in the exoskeletal robotic training group for knee flexion torque, walking distance, and quality of life. In addition, our study showed that the exoskeletal robotic training was the only predictive factor for post-stroke patients having better improvement in both the 6 min walk test and the total score of SF-12.

Robotic exoskeletal devices must receive instructions to assist the movement and function of the weak limbs of patients. Commonly adopted strategies include sensing the patient’s intended trivial movements of limbs and magnifying them, or receiving signals of surface electromyography for selected muscles and assisting their direction of force [26]. These strategies can be severely confounded by spasticity, a common neurological sequela for stroke patients. If spasticity exists in both agonist and antagonist muscles, the robotic devices cannot receive accurate orders and thus are not able to provide meaningful assistance. Another way is to directly give commands to the control system of the robotics, either by therapists’ manual control or via brain–computer interface devices. Although the brain–computer interfaces are emerging devices under rapid development, there are still many obstacles for clinical use, such as the noise filtering of signals from electroencephalography, and precisely measuring the signals from the cortex controlling the lower limbs, which lies just between the hemispheres of the brain [27]. In our study, we used the FREE Walk exoskeletal device (FREE Bionics, Hsin-Chu, Taiwan) to perform robotic-assisted gait training for our participants, which is operated by direct manual control by therapists during training. It avoids the interference of post-stroke spasticity, allows trunk balance training during assisted walking, and provides data regarding the assistive percentage in gait cycles. This device can even be applied in cases of patients with completely paralyzed lower limbs, allowing the patients to regain walking ability. In our study, we only recruited stroke patients who took more than 20 s to complete the Timed Up-and-Go test, since the FREE Walk exoskeletal device can provide very little assistance for those who can already walk very well. On the other hand, patients who could not finish the Timed Up-and-Go test were also excluded, because the time taken in this test is one of our outcome measures. If we recruited such patients, their pre-training data would be unobtainable, even though these patients can still benefit from the FREE Walk exoskeletal device.

Our participants showed significant improvement in both knee flexion and extension strengths after robotic training. A previous study by Gil-Castillo et al. found a trend of increased strength in all joints of the lower limb, but with statistical significance in only hip abduction and knee flexion muscle strength [28]. Another study by Irene et al. revealed that the muscle strength of lower limbs could not be improved through merely performing gait training with an end-effector robotic system. The strength could only be improved through combined balance training [29]. Our study, using an exoskeletal type of robotic device, provided combined gait and balance training for our patients. Therefore, our study results are consistent with Irene’s report. A large systematic review in 2021, involving 1219 studies and 10 treatment guidelines, concluded that the robotic exoskeleton for the lower extremities improves the lower extremity function of stroke patients, including gait and leg muscle strength [19]. These findings are in line with our present results.

Compared with the experimental group, the control group showed no improvement regarding torque of knee flexion and extension at 120°/s. This finding is possibly due to the fact that, without the help of the robotic exoskeleton, the restoration of fast movement of the knee is slower. A previous study that analyzed and compared the paretic and non-paretic legs of stroke survivors discovered that a greater percentage of type II fast-twitch white muscle fibers had appeared in the paretic legs [30]. Therefore, we speculate that the potential of strength recovery in fast-twitch muscles is smaller due to the pre-existing higher percentages of type II muscle fiber before training. Compared to traditional rehabilitation programs, adding the robotic training could bring significant strength recovery in these fast-twitch fibers.

In the between-group comparison of training improvements, our study showed a greater effect on the walking speed and knee flexion torque at 60°/s in the robotic group. A study by Choi reported that robotic rehabilitation training had better improvement in the 10 m walk and the Timed Up-and-Go tests compared with a control group [31]. On the contrary, our study revealed only a significantly longer 6 MWT distance and it failed to show more improvement in the Timed Up-and-Go tests for the robotic group. The reason probably lies in the fact that the intensity and duration of Choi’s study were much greater than ours, comprising robotic training programs five times a week for six weeks. On the contrary, our robotic programs were executed only three times a week for four weeks. Regarding muscle strength, Kim et al. used the Motricity Index-Lower to evaluate the muscle strength of lower limbs, and they found that robotic exoskeleton training increases muscle strength more as compared with the general traditional rehabilitation training [32]. While the Motricity Index-Lower evaluates the composite muscle strength of hip, knee, and ankle, our study only took knee strength into consideration. Our study further used an isokinetic strength test to quantify the amount of strength improvement, and our results are consistent with the literature.

In our research, the LDL value was shown to be the only independent predictor regarding improvement on the Timed Up-and-Go test. A previous study by Yang et al. revealed that a significant decrease in the oxidized LDL level predicts a more favorable functional outcome, defined as a modified Rankin Scale grade 3 to 6 at 90 days after stroke [33]. The possible mechanism may be related to a higher degree of endothelial dysfunction and circulating pro-inflammatory cytokines brought by higher LDL [34], which may, in turn, prohibit neuronal regeneration and reinnervation. However, the specific underlying mechanism is still unclear, and more studies are needed.

Regarding the patient’s quality of life, our results were not consistent with those of a previous study by Dundar et al., in which they used the SF-36 to measure the quality of life [35]. The improvement in all SF-36 sub-items in the robot training group showed significant differences compared with the general training group, whereas in our present study we only found significant improvement in the mental subdomain score and the total score. The study by Dundar et al. was retrospective, and the number of patients was larger at 107. The results of the improvement in the physical subdomain scores of the SF-12 might have been different if we had more participants in our study. However, the total score on the SF-12 had the largest effect size in the between-group analysis, signifying that the better quality of life brought by robotic training was already the most significant among all of the outcome measures for our patients.

In the past, adverse events have been reported during robotic gait training, such as soft tissue injury, including skin irritation, abrasion, and bruising, and musculoskeletal injury, including tendinitis, tibia fracture, muscle ache, and malleolar pain [36]. Because the robotic exoskeletal device cannot maintain balance by itself, preventing falling accidents during gait training is of paramount importance. A well-trained assistant can stabilize the exoskeletal device as well as the patient during walking, which is a prerequisite for safety. Furthermore, fine adjustment of the length of each part of the exoskeletal device to perfectly fit the pelvis and legs is also crucial to prevent soft tissue injury. In our study, no participants were harmed during the exoskeletal gait training program.

The strength of our study is that we used both subjective and objective tools as our outcome measures, and the isokinetic dynamometer was adopted to better quantify the degree of muscle strength improvement. Furthermore, we took all the baseline characteristics into consideration when determining predictive factors of significant improvement in post-stroke patients. Despite such strengths, our study still has at least three limitations, which could influence the presented results. First, we had no placebo control group, and we therefore cannot rule out that the improvements in both groups were related to the natural history of stroke recovery. Nevertheless, it is unethical to leave post-stroke patients untreated with a rehabilitation program. Second, our study protocol was not blinded to either participants or assessors. The primary outcome measures including peak torque of lower limbs, the distance covered in the 6 min walk test, and the time taken in the Timed Up-and-Go test were generally objective. Such objectivity should not be affected by our non-blinded design. On the other hand, the SF-12 questionnaire is subjective, and that may be influenced by our study design. Third, the post-rehabilitation evaluation was conducted right after the last training session, so the long-term effects of exoskeletal robotic-assisted training were not revealed. Therefore, future studies regarding the long-term outcomes are warranted.

## 5. Conclusions

Robotic-exoskeleton-assisted gait rehabilitation resulted in better lower limb strength, walking speed, and quality of life in hemiparetic stroke patients whose onset had been within 6 months. Receiving robotic rehabilitation was also proven to be a significant predictor of better improvement in a 6 min walk test and the SF-12 through our study. Therefore, post-stroke patients with gait disturbance, whose Timed Up-and-Go test takes more than 20 s to complete, are recommended to receive robotic gait training if the device is available and the expenditure is affordable. If the device is not available, traditional rehabilitation programs could still bring benefits for stroke patients.

## Figures and Tables

**Figure 1 bioengineering-10-00585-f001:**
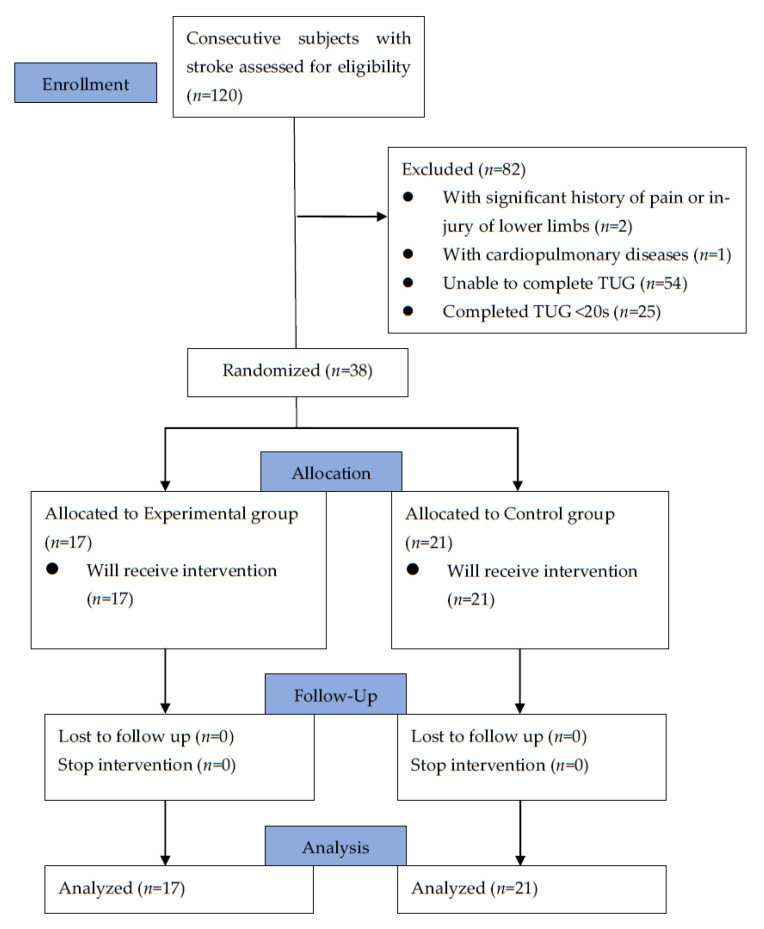
CONSORT flow diagram of subject recruitment and retention.

**Figure 2 bioengineering-10-00585-f002:**
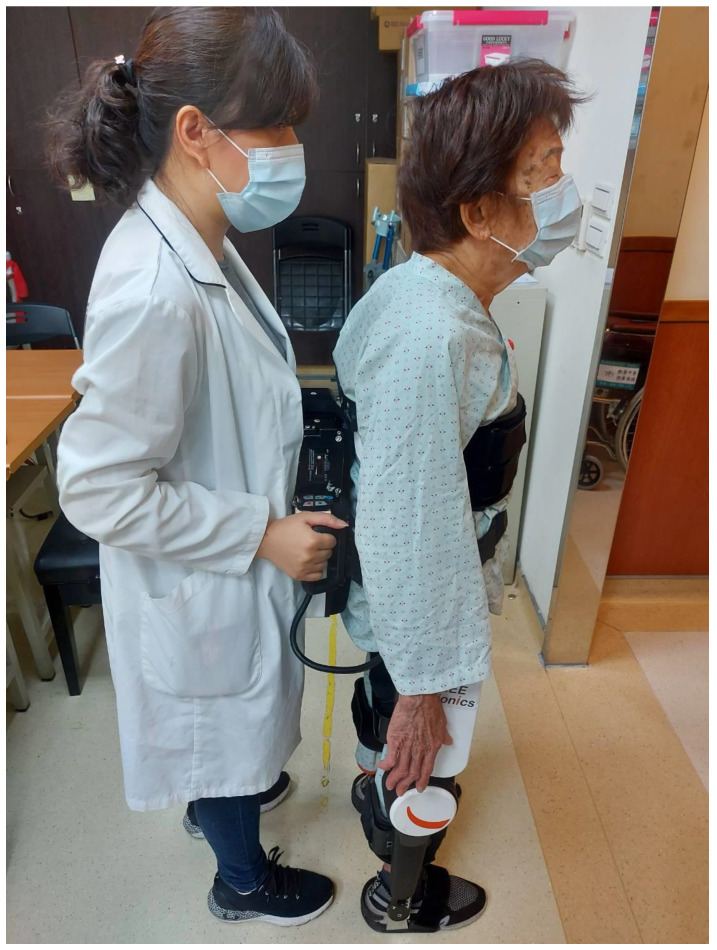
The robotic-exoskeleton-assisted gait training.

**Table 1 bioengineering-10-00585-t001:** Demographic and clinical characteristics of participants.

	Experimental Group (*n* = 17)		Control Group (*n* = 21)		*p*-Value
	Median	IQR	Median	IQR	
Age (yr)	68.0	(56.5–74.5)	70.0	(63.5–74.5)	0.527
Sex (*n*,%)					0.444
Male	11	(64.71%)	11	(52.38%)	
Female	6	(35.29%)	10	(47.62%)	
Height (cm)	158.00	(155.50–168.25)	160.00	(158.10–165.50)	0.499
Weight (kg)	69.40	(57.65–70.95)	72.00	(60.60–77.30)	0.122
BMI	24.60	(21.90–27.35)	26.70	(23.56–29.48)	0.082
NIHSS score	6	(3–10)	7	(4.5–11.5)	0.594
HbA1C (%)	6.30	(5.60–7.15)	6.60	(6.20–7.55)	0.181
LDL (mg/dL)	105.00	(78.00–135.00)	121.00	(100.50–179.00)	0.072
Homocysteine (μmol/L)	11.05	(7.99–12.52)	11.00	(9.50–16.00)	0.310
Stroke type (*n*,%)					1.000
Hemorrhagic	5	(29.41%)	6	(28.57%)	
Ischemic	12	(70.59%)	15	(71.43%)	
Stroke site (*n*,%)					0.407
Left	6	(35.29%)	11	(52.38%)	
Right	10	(58.82%)	10	(47.62%)	
Bilateral	1	(5.88%)	0	(0%)	
Pre-rehabilitation data					
Ext. 60 (N⋅m)	10.00	(4.35–20.20)	10.00	(8.00–13.50)	0.755
Flex. 60 (N⋅m)	9.10	(6.50–10.90)	11.00	(8.50–14.50)	0.095
Ext. 120 (N⋅m)	9.90	(3.95–14.25)	12.00	(9.00–14.50)	0.176
Flex. 120 (N⋅m)	11.60	(8.00–15.00)	15.00	(10.50–17.50)	0.066
6MWT (m)	10.00	(6.00–37.50)	10.00	(8.50–12.00)	0.867
TUG (s)	61.33	(50.00–115.86)	57.07	(41.16–70.29)	0.107
SF-12	108.50	(71.63–114.78)	106.90	(100.61–112.24)	0.685

IQR = interquartile range; BMI = body mass index; LDL = low-density lipoprotein; Ext. 60 = extension torque at 60 degrees per second; Flex. 60 = flexion torque at 60 degrees per second; Ext. 120 = extension torque at 120 degrees per second; Flex. 120 = flexion torque at 120 degrees per second; 6 MWT = 6 min walk test; TUG = Timed Up-and-Go test; SF-12 = 12-Item Short Form Survey.

**Table 2 bioengineering-10-00585-t002:** Within-group post-rehabilitation changes of both groups.

	Pre-Rehabilitation		Post-Rehabilitation		*p*-Value
	Median	IQR	Median	IQR	
**Experimental group**					
Ext. 60 (N⋅m)	10.00	(4.35–20.20)	19.00	(13.20–37.60)	0.001 **
Flex. 60 (N⋅m)	9.10	(6.50–10.90)	12.00	(9.25–16.50)	0.001 **
Ext. 120 (N⋅m)	9.90	(3.95–14.25)	13.70	(8.00–23.85)	0.010 *
Flex. 120 (N⋅m)	11.60	(8.00–15.00)	13.80	(12.10–20.25)	0.001 **
6 MWT (m)	10.00	(6.00–37.50)	33.00	(28.50–60.00)	0.007 **
TUG (s)	61.33	(50.00–115.86)	44.18	(33.65–83.72)	0.006 **
SF-12 physical domain	52.49	(47.52–54.98)	54.59	(50.34–60.45)	0.266
SF-12 mental domain	65.09	(59.13–112.92)	58.09	(51.86–70.02)	0.136
SF-12 total score	108.50	(71.63–114.78)	117.71	(111.88–121.14)	0.005 **
**Control group**					
Ext. 60 (N⋅m)	10.00	(8.00–13.50)	14.00	(9.00–22.50)	0.001 **
Flex. 60 (N⋅m)	11.00	(8.50–14.50)	14.00	(10.00–16.00)	0.012 *
Ext. 120 (N⋅m)	12.00	(9.00–14.50)	14.00	(9.00–16.00)	0.140
Flex. 120 (N⋅m)	15.00	(10.50–17.50)	16.00	(9.50–23.50)	0.120
6 MWT (m)	10.00	(8.50–12.00)	9.00	(8.00–12.00)	0.392
TUG (s)	57.07	(41.16–70.29)	55.89	(35.76–61.32)	0.038 *
SF-12 physical domain	53.28	(47.22–55.26)	50.34	(42.58–53.51)	0.017 *
SF-12 mental domain	54.53	(51.78–60.06)	55.34	(49.69–58.82)	0.259
SF-12 total score	106.90	(100.61–112.24)	103.30	(96.85–110.15)	0.016 *

IQR = interquartile range; Ext. 60 = extension torque at 60 degrees per second; Flex. 60 = flexion torque at 60 degrees per second; Ext. 120 = extension torque at 120 degrees per second; Flex. 120 = flexion torque at 120 degrees per second; 6 MWT = 6 min walk test; TUG = Timed Up-and-Go test; SF-12 = 12-Item Short Form Survey; * *p* < 0.05; ** *p* < 0.01.

**Table 3 bioengineering-10-00585-t003:** Between-group comparison of post-rehabilitation improvement.

	Experimental Group (*n* = 17)		Control Group (*n* = 21)		*p*-Value		Mean Difference
	Mean	SD	Mean	SD		Effect Size (d)	
Ext. 60 (N⋅m)	9.48	9.18	4.38	6.89	0.068	0.62	5.1
Flex. 60 (N⋅m)	4.38	3.61	2.05	3.4	0.048 *	0.67	2.33
Ext. 120 (N⋅m)	4.14	6.36	1.67	4.54	0.171	0.46	2.47
Flex. 120 (N⋅m)	3.92	3.83	2.76	8.3	0.598	0.17	1.16
6 MWT (m)	25.59	49.3	−0.19	1.08	0.047 *	0.7	25.78
TUG (s)	−22.77	38.07	−3.4	5.75	0.054	0.68	−19.37
SF-12 physical domain	4.22	13.56	−2.54	5.08	0.066	0.63	6.75
SF-12 mental domain	−20.98	33.12	−0.95	4.63	0.025 *	0.8	−20.03
SF-12 total score	19.46	25.97	−3.48	6.44	0.002 **	1.16	22.93

SD = standard deviation; Ext. 60 = extension torque at 60 degrees per second; Flex. 60 = flexion torque at 60 degrees per second; Ext. 120 = extension torque at 120 degrees per second; Flex. 120 = flexion torque at 120 degrees per second; 6 MWT = 6 min walk test; TUG = Timed Up-and-Go test; SF-12 = 12-Item Short Form Survey; * *p* <0.05; ** *p* <0.01.

**Table 4 bioengineering-10-00585-t004:** Univariate logistic analyses to determine predictors of improvement on 6 min walk test and Timed Up-and-Go test.

	6 Min Walk Test			Timed Up-and-Go Test		
	OR	95% CI	*p*-Value	OR	95% CI	*p*-Value
Training type						
Experimental	31.87	(5.09–199.48)	<0.001 **	0.34	(0.09–1.27)	0.107
Control	1.00			1.00		
Age (yr)	0.99	(0.93–1.05)	0.678	0.99	(0.93–1.04)	0.615
Sex						
Male	1.00			1.00		
Female	0.65	(0.18–2.37)	0.512	0.42	(0.11–1.56)	0.193
BMI	0.85	(0.70–1.03)	0.092	1.09	(0.90–1.30)	0.380
NIHSS score	1.03	(0.91–1.18)	0.614	1.02	(0.89–1.16)	0.813
HbA1C (%)	0.77	(0.42–1.44)	0.417	1.11	(0.61–2.02)	0.738
LDL (mg/dL)	0.99	(0.97–1.10)	0.132	1.02	(1.00–1.04)	0.033 *
Homocysteine (μmol/L)	0.93	(0.81–1.08)	0.347	1.04	(0.90–1.20)	0.583
Stroke type						
Hemorrhagic	1.00			1.00		
Ischemic	1.29	(0.32–5.28)	0.721	2.19	(0.52–9.27)	0.288
Stroke site						
Left	1.00			1.00		
Right	1.90	(0.52–6.96)	0.330	0.81	(0.22–2.91)	0.744
Pre-rehabilitation data						
Ext. 60 (N⋅m)	1.03	(0.97–1.10)	0.305	1.02	(0.96–1.08)	0.514
Flex. 60 (N⋅m)	0.97	(0.86–1.09)	0.592	1.01	(0.90–1.13)	0.874
Ext. 120 (N⋅m)	1.03	(0.94–1.12)	0.511	1.03	(0.95–1.13)	0.471
Flex. 120 (N⋅m)	1.00	(0.92–1.09)	0.989	0.99	(0.91–1.08)	0.848
6 MWT (m)	0.99	(0.96–1.03)	0.756	1.01	(0.97–1.05)	0.620
TUG (s)	1.02	(1.00–1.04)	0.134	0.99	(0.97–1.01)	0.271
SF-12 physical domain	1.03	(0.91–1.17)	0.603	0.96	(0.85–1.09)	0.540
SF-12 mental domain	1.03	(1.00–1.07)	0.078	1.00	(0.97–1.02)	0.760
SF-12 total score	0.98	(0.94–1.02)	0.296	0.98	(0.95–1.02)	0.430

OR = odds ratio; CI = confidence interval; BMI = body mass index; LDL = low-density lipoprotein; Ext. 60 = extension torque at 60 degrees per second; Flex. 60 = flexion torque at 60 degrees per second; Ext. 120 = extension torque at 120 degrees per second; Flex. 120 = flexion torque at 120 degrees per second; 6 MWT = 6 min walk test; TUG = Timed Up-and-Go test; SF-12 = 12-Item Short Form Survey; * *p* < 0.05 ** *p* < 0.01.

**Table 5 bioengineering-10-00585-t005:** Univariate and multivariate logistic analyses for predictors of improvement on SF-12 score.

	Univariate Analysis			Multivariate Analysis		
	OR	95% CI	*p*-Value	OR	95% CI	*p*-Value
Training type						
Experimental	14.93	(3.01–74.04)	0.001 **	32.09	(2.82–364.82)	0.005 **
Control	1.00					
Age (yr)	1.03	(0.97–1.09)	0.378			
Sex						
Male	1.00					
Female	0.65	(0.18–2.37)	0.512			
BMI	0.87	(0.72–1.05)	0.156			
NIHSS score	0.88	(0.76–1.02)	0.097			
HbA1C (%)	0.74	(0.76–1.02)	0.353			
LDL (mg/dL)	0.99	(0.98–1.00)	0.184			
Homocysteine (μmol/L)	1.07	(0.93–1.24)	0.341			
Stroke type						
Hemorrhagic	1.00					
Ischemic	0.77	(0.19–3.16)	0.721			
Stroke site						
Left	1.00					
Right	2.98	(0.93–1.24)	0.107			
Pre-rehabilitation data						
Ext. 60 (N⋅m)	1.06	(0.99–1.14)	0.108			
Flex. 60 (N⋅m)	1.10	(0.95–1.27)	0.198			
Ext. 120 (N⋅m)	1.10	(0.98–1.23)	0.103			
Flex. 120 (N⋅m)	1.00	(0.92–1.10)	0.926			
6 MWT (m)	1.03	(0.98–1.07)	0.284			
TUG (s)	1.01	(0.99–1.03)	0.358			
SF-12 physical domain	0.88	(0.77–1.01)	0.076			
SF-12 mental domain	1.08	(1.00–1.17)	0.048 *			
SF-12 total score	0.95	(0.90–0.997)	0.039 *	0.93	(0.85–1.02)	0.118

OR = odds ratio; CI = confidence interval; BMI = body mass index; LDL = low-density lipoprotein; Ext. 60 = extension torque at 60 degrees per second; Flex. 60 = flexion torque at 60 degrees per second; Ext. 120 = extension torque at 120 degrees per second; Flex. 120 = flexion torque at 120 degrees per second; 6 MWT = 6 min walk test; TUG = Timed Up-and-Go test; SF-12 = 12-Item Short Form Survey; * *p* < 0.05; ** *p* < 0.01.

## Data Availability

The raw data of this study is not to be provided according to the rule of the ethics committee.

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
