# Peer review of "Therapeutic Effects of Robotic-Exoskeleton-Assisted Gait Rehabilitation and Predictive Factors of Significant Improvements in Stroke Patients: A Randomized Controlled Trial"

_bioengineering, 2023, doi:10.3390/bioengineering10050585_

Round 1

Reviewer 1 Report

The authors conducted a randomized controlled trial to examine the effects of robotic exoskeleton-assisted gait training on limb muscle strength and functioning in post-stroke patients. Thirty-eight post-stroke hemiparetic patients were randomized to undergo robotic exoskeleton-assisted gait or regular rehabilitation for four weeks. The authors showed that those receiving robotic exoskeleton-assisted gait rehabilitation showed better improvements in multiple outcome measures, in particular 6-minute walk test and SF-12. The results are interesting. There are some comments.

Comments:

1.      Materials and Methods (Line 95 on Page 4): “Patients were assigned into two groups using a simple randomization method.” However, it is unclear how randomization was conducted. I would suggest a more detailed description of the method (for instance, the computer program used in randomization, the mechanism of allocation concealment, who generated the sequence, who enrolled participants, and who assigned participants to treatment arms, etc.). Moreover, please specify the allocation ratio. If blocking was used, it (as well as the block size) should be described.

2.      Materials and Methods (Line 97-100 on Page 4): “The control group received regular post-stroke rehabilitation programs 5 days a week, and the experimental group received robotic exoskeleton-assisted gait rehabilitation.” “As for robotic rehabilitation, we used the FREE Walk exoskeletal device (FREE Bionics, Hsin-Chu, Taiwan).” I would suggest describing how the robotic exoskeleton-assisted gait rehabilitation and regular post-stroke rehabilitation programs were administered in more detail.

3.      Materials and Methods (Line 103-106 on Page 4): “The outcome measures of our study included the 6 minute walk test, Timed Up-and-Go test, quadriceps isokinetic muscle strength test using an isokinetic dynamometer -, and the 12-item short form survey (SF-12).” Were all these the pre-specified primary outcome measures? Or one was the primary outcome measure, and the others were the secondary outcomes? In addition, according to the analysis conducted, the primary and secondary outcomes were changes in these measures instead of the measures themselves. A more specific description is recommended.

4.      Results (Table 3): Please present the effect size (and 95% confidence interval) of each measure (for instance, median difference of change). Also, I suggest conducting a sensitivity analysis in which the absolute change, instead of the relative change, was analyzed.

5.      Discussion: As shown in Table 3, improvement in TUG was significantly greater in those receiving robotic exoskeleton-assisted gait training than in those receiving regular rehabilitation. However, analysis in Table 4-1 showed that receiving robotic exoskeleton-assisted gait training (vs. control) was not associated with improvement in TUG with OR of 0.34 (95% CI, 0.09-1.27). How would these conflicting results be explained? An in-depth discussion is recommended.

6.      Discussion: Would robotic exoskeleton-assisted gait training have harmful or untended effects? A discussion of this issue is suggested.

Author Response

Thanks a lot for the reviewer’s detailed examination of our paper and the valuable comments. Many good points have been raised, and this has helped us to present our study more clearly. The manuscript has been revised based on these suggestions. Our responses to the reviewer’s comments are given below on a point-by-point basis.

  1. Common randomization methods used in clinical studies include simple randomization, block randomization, stratified randomization, and covariate adaptive randomization.1 In our study, we simply adopted the simple randomization method by tossing a coin, which is involved in neither the computer sequence generating nor the blocking allocation. We’ve revised our manuscript as following to make it clearer to the readers: “Specifically, patients fulfilling our study criteria were recruited by doctors during inpatient or outpatient visiting, and then the study assistant decided which group the participants would join by tossing a coin.”
  2. We agree that the original version of the content of rehabilitation program is too simplified, and we’ve revised it as following: “The regular post-stroke rehabilitation programs were designed and performed by an experienced physical therapist, including facilitation techniques for the prominently weak motion of the affected limbs, strengthening exercise for specific muscle groups needed in daily activities, stretching exercise for the spastic muscle groups, trunk balance training, ambulation training, posture education, positioning advice, orthotic devices fitting, and functional training targeting at improving activities of daily life. As for robotic rehabilitation, we used the FREE Walk exoskeletal device (FREE Bionics, Hsin-Chu, Taiwan) to specifically address the issue of walking, standing up and sitting down. Participants practiced the three kinds of movement repeatedly wearing the exoskeletal device on their lower limbs, with a study assistant standing behind to help the patient to maintain balance and control the training programs, as shown in Figure 2.”
  3. As described in the 2.1. Participants and Trial Design section, we used the preliminary data of the improvement of the time taken in the timed up-and-go test as the primary assessed outcome to perform sample size calculation. Therefore, the timed up-and-go test was the primary outcome measure in our study, and others were the secondary outcomes. We’ve revised the manuscript as following: “The primary outcome measure of our study was the improvement of Timed Up-and-Go test, and the secondary outcome measures included the improvement of the 6 minute walk test, quadriceps isokinetic muscle strength test using an isokinetic dynamometer (Biodex Multi-Joint System 3; Biodex Medical, New York, USA), and the 12-item short form survey (SF-12). These outcome measurements were conducted twice for each participant: once before and once after the 4-week post-stroke rehabilitation program. The purpose was to assess and compare the improvements of walking speed, function of lower limbs, quadriceps strength and quality of life in the two groups.”
  4. Because our data was not normally distributed during Kolmogorov-Smirnov test, we had performed Mann-Whitney U test to detect if there’s significant inter-group differences. However, as required by the reviewer, independent sample T test was performed to calculate the effect size and mean difference of change, which generated a completely different result in the p value. Therefore, we revised our manuscript accordingly.
  5. Indeed, the original results obtained from Mann-Whitney U test, which generated a statistically significant improvement in TUG in the robotic group, cannot explain the result of further logistic regression. However, through the reviewer’s valuable suggestion, the independent sample T test showed no significant differences in the improvements of TUG between the two groups, which make the results of our study consistent between each other.
  6. The issue of safety in robotic gait training deserves discussion, and we revised our manuscript as following: “In the past, adverse events were reported during robotic gait training, such as soft tissue injury, including skin irritation, abrasion, and bruising, and musculoskeletal injury, including tendinitis, tibia fracture, muscle ache, and malleolar pain2. Because the robotic exoskeletal device cannot maintain balance by itself, preventing falling accidents during gait training is of paramount importance. A well-trained assistant can stabilize the exoskeletal device as well as the patient during walking, which is a prerequisite for safety. Furthermore, fine adjustment of the exoskeletal device to perfectly fit the pelvis and legs is also crucial to prevent soft tissue injury. In our study, no participants were harmed during the exoskeletal gait training program.”

References

  1. Suresh K. An overview of randomization techniques: An unbiased assessment of outcome in clinical research. Journal of human reproductive sciences 2011;4(1):8-11.
  2. Bessler J, Prange-Lasonder GB, Schaake L, Saenz JF, Bidard C, Fassi I et al. Safety Assessment of Rehabilitation Robots: A Review Identifying Safety Skills and Current Knowledge Gaps. Frontiers in robotics and AI 2021;8:602878.

Reviewer 2 Report

Thanks to the authors for this manuscript, which is of great interest in the field of neurorehabilitation. However, there are several issues that should be expanded and/or modified. Below I point out the indications derived from the review carried out.

Interventions

The authors should expand and detail the type of intervention, for example, what they refer to as regular post-stroke control group, what professionals as each one... aims of the intervention... this section can be greatly improved.

assessments

Describe the tools used such as scores, number of items...

What was the reason for using the SF-12, and not other tools such as WHOQOL-BREF, CAVIDACE...

Was the psychological situation not assessed? In this group it is something of an essential nature.

results

Sex include nonbinary

discussion

224-226 Have these results been found in the literature? There is a lack of debate between what the authors found and the previous literature in the first paragraph.

270 how do you know they could become significant? This is totally subjective, I don't think it should appear.

In limitations, the scarcity of participants, type of sampling... the cost of the interventions differs depending on the group? It is not explained in the text.

conclusion

Motor performance where is it valued?

Author Response

Thanks a lot for the reviewer’s detailed examination of our paper and the valuable comments. Many good points have been raised, and this has helped us to present our study more clearly. The manuscript has been revised based on these suggestions. Our responses to the reviewer’s comments are given below on a point-by-point basis.

  1. We agree that the original version of this paragraph was too simplified, and we’ve revised it as following: “The regular post-stroke rehabilitation programs were designed and performed by an experienced physical therapist, including facilitation techniques for the prominently weak motion of the affected limbs, strengthening exercise for specific muscle groups needed in daily activities, stretching exercise for the spastic muscle groups, trunk balance training, ambulation training, posture education, positioning advice, orthotic devices fitting, and functional training targeting at improving activities of daily life. As for robotic rehabilitation, we used the FREE Walk exoskeletal device (FREE Bionics, Hsin-Chu, Taiwan) to specifically address the issue of walking, standing up and sitting down. Participants practiced the three kinds of movement repeatedly wearing the exoskeletal device on their lower limbs, with a study assistant standing behind to help the patient to maintain balance and control the training programs.”
  2. Most outcome measures of our study were objective, such as TUG, 6MWT, and isokinetic torque. The only subjective questionnaire we adopted was SF-12, and we’ve explained the content of it in detail in our paragraph. The reason why we chose SF-12 instead of WHOQOL-BREF(28 questions) and CAVIDACE(64 questions) based primarily on its simplicity to administer and proven reliability in stroke patients without loss of information comparing to SF-36.
  3. About the psychological conditions, we excluded those with cognitive impairment during recruitment, and then we assessed it by the mental subdomain of SF-12. Through further review of the medical charts, none of our participants had records of psychological problems.
  4. We agree that there might be nonbinary sex. However, all of the participants have definite gender identity as either male or female in our study. We’ve added this information in “3.1. Demographic and Clinical Characteristics of participants” section.
  5. In the past literatures, there are indeed some reports regarding to the effect of robotic rehabilitation on lower limb strength, walking speed, timed up and go test, and quality of life, just as in our reference 14, 20, 21, 22, 23, and 25. We compared our results with these references in the later sections of discussion paragraphs.
  6. We agree with the reviewer’s opinion, and the paragraph was revised as following: “The results of the improvements in the physical subdomain scores of SF-12 may be different if we could have more participants in our study.”
  7. About the number of participants, we did performed sample size calculation according to our preliminary results, as shown in the last part of 2.1. Participants and Trial Design. About the cost of exoskeletal training, our participants did not pay anything for this study. About the type of sampling and randomization, we agree with the reviewer, and we’ve revised our paragraph to make it clearer as following “Specifically, patients fulfilling our study criteria were recruited by doctors during inpatient or outpatient visiting, and then the study assistant decided which group the participants would join by tossing a coin” in section 2.2 interventions.
  8. We agree with the reviewer, and the word “motor performance” was removed in the paragraph.

Reviewer 3 Report

This study verifies the excellent effect of robotic exoskeleton-assisted gait training in stroke rehabilitation and identifies the main factors of improvement in post-stroke patients.

1. The design of the experiment was well done, and the results are well documented to support the conclusions.

2. However, all results are organized using tables, but it is necessary to use graphs to give visual effects.

3. If possible, it is necessary to include in the discussion or conclusion a consideration of which characteristics of the program using robotic exoskeleton compared to general rehabilitation programs influenced the presented results.

Author Response

Thanks a lot for the reviewer’s detailed examination of our paper and the valuable comments, and this has helped us to present our study more clearly. The manuscript has been revised based on these suggestions. Our responses to the reviewer’s comments are given below on a point-by-point basis.

  1. Thanks a lot for the reviewer’s comment on the design of our study.
  2. We agree with the reviewer’s valuable suggestion, and we’ve made a graphical abstract for our study as attached file. Furthermore, Figure 2 was also added to present the actual condition of robotic training on site.
  3. Thanks for the reviewer’s comment. Our study program did have some characteristics that may influence the presented results, as shown in the section of study strength and limitations. We’ve revised our manuscript to better address this point of view as following: “The strength of our study is that we used both subjective and objective tools as our outcome measures, and the isokinetic dynamometer was adopted to better quantify the degree of muscle strength improvement. Furthermore, we took all the baseline characteristics into consideration when determining predictive factors of significant improvement in post-stroke patients. Despite such strengths, our study still has at least 3 limitations, which could influence the presented results.”
